# A Creative Approach to Knowledge Translation: The Use of Short Animated Film to Share Stories of Refugees and Mental Health

**DOI:** 10.3390/ijerph191811468

**Published:** 2022-09-12

**Authors:** Katherine M. Boydell, Joseph Croguennec

**Affiliations:** 1Black Dog Institute, Sydney 2034, Australia; 2Department of Psychiatry and Mental Health, University of New South Wales, Sydney 2034, Australia

**Keywords:** knowledge translation, animation, film, mental health, refugee, arts-based methods

## Abstract

This study used animated film to translate narratives of refugees and mental health into accessible material aimed at enhancing empathy and understanding. It focuses on the use of short animated films in series one and two of the Woven Threads catalogue. Series one shared moments of hope in a refugee’s journey, whilst series two focused on people living with mental health challenges. This research was designed to understand the responses to viewing for people who watch these animations. A mixed-method design was used via an online Qualtrics platform that asked respondents to view two short animated films, one from the refugee series and one from the mental health series. 364 members of the general public viewed and responded to the refugee film and 275 responded to the mental health film. The platform collected both quantitative and qualitative data. Survey responses indicated that the majority of viewers found the films challenged public misconceptions about refugees and individuals with mental health challenges and left them with a feeling of hopefulness. Qualitative narratives were organised into one superordinate theme: the power of film as a knowledge translation strategy, with four subthemes: (i) changing perceptions and inspiring empathy, (ii) enhancing literacy, (iii) highlighting the power of storytelling, and (iv) encouraging hope and a sense of belonging. The use of short animated film as a knowledge translation strategy can enhance our understanding, promote deep reflection, increase empathy and has the potential to lead to social change.

## 1. Introduction

Translating health research evidence into practice is challenging and, consequently, the field of knowledge translation (KT) has bourgeoned. Knowledge translation encompasses the research cycle from identification of the issue or problem, the creation of new knowledge through to dissemination of results. In academia, scholars typically disseminate their research findings via scientific presentations and peer reviewed journal articles [1]. However, these knowledge translation strategies often fail to reach the stakeholders who need to know–clinicians, policy makers, health care consumers, their families and the public [1,2]. This failure in reach can be due to inability to access peer-reviewed journal articles, lack of time to read them, and lack of familiarity with the language used [1].

In the field of health and social care, there is a strong need to make sense of individual health experiences via personal narratives [3]. Consequently, the power of stories to communicate what it is like to experience a healthcare issue needs to be highlighted. With the exponential increase in interest in knowledge translation, there are a plethora of types of visual representation of evidence-based experiential narratives that have been the topic of study, including digital storytelling [4,5], research-based theatre [6] and story-based e-books [7].

Underpinning the work of Archibald and her colleagues [3] is the underscoring of the benefits of storytelling, as an essential component of what it is to be human, our expression of selfhood, and as a way to both learn and relate to the experiences of others. The impact of stories in health and social care have been highlighted by Greenhalgh and her colleagues in the form of narrative medicine [8]. Combining the power of stories with visual representations to create and communicate, and to engage with evidence-based narratives has been of growing interest to KT, health communication, and public health scholars [3,7].

Personalized knowledge translation strategies are required to expediate learning and accelerate action in community-based contexts. Evidence-informed documentary style videos represent one such knowledge translation tool. For example, Ottoni et al. [9] video had a positive impact on learning environment, participant understanding and actions. Their findings provide support for the claim that innovative strategies such as film and video can successfully educate, encourage and activate older adults with respect to their independence, physical activity and social connectedness.

Among scholars in the public health realm, there is developing attention to the use of film methods due to their ability to emphasize nuances in practices, highlight emotions, engage difficult-to-reach groups, and advocate for social change. In spite of this growing interest, there is a paucity of knowledge regarding the strengths and challenges associated with using film methods in health and social care. A recent review synthesizes peer-reviewed, public health research studies that apply film methods, and describes opportunities and challenges [10]. Of the 3431 identified articles, 20 met the inclusion criteria. Fifteen different film methods were identified which offer numerous methodological strengths of using film, including the provision of richly detailed description, capturing cultural norms, enhancing the comfort level of participation, empowering participants, and using film for advocacy. Other research supports these findings. For example, Coughlan et al. [11] aimed to develop engaging and accessible public mental health animations for young people based on their thematic findings that anxiety, depression, feeling different, loneliness, and being bullied were commonly experienced by young people. They demonstrated that online animations provided an accessible way to translate empirical research findings into meaningful public health products. The films offer an economical way to provide targeted online information about mental health, coping, and help-seeking to young people. Vaughn et al. [2] described their use of animated film as an innovative approach to disseminate their research findings on youth violence prevention to their targeted communities via digital animation. This paper’s aim was to extend the extant literature in the field and to focus on the response of the general public who viewed one of two short animated films in the Woven Threads series, described below, that focused on refugees [Hisham’s story] and mental illness [Olive’s story] respectively.

## 2. Materials and Methods

### 2.1. The Woven Threads Series

The Woven Threads film series was created by Michi Marosszeky and Paul Sullivan, founders of Woven Threads Productions, with the support of an animator and other artists. The films were created by professionals, all of whom have a great deal of experience in film and television and who wanted to use those skills to create work that could have a social impact. Woven Threads is focused on telling stories about our shared humanity. These stories focus on people and their courage, resilience, gratitude, compassion and hope. People and their lived experiences are central to the stories; empathy is a key consideration as it represents the key to understanding. Understanding, in turn, is the key to social change [12].

The first series on refugees was the chosen because the creator of the series is from a refugee family and wanted to change the dialogue around refugees and find a way for the community to move past the labels and connect with the humanity of a refugee. During the time of making this series, Michi and Paul experienced a variety of issues with their daughters and mental health and they realised that the experience was quite isolating for them and their children. This then led to the next topic being mental health and the further development of Woven Threads to become an ongoing platform for change, through people sharing their lived experiences in a way that is relatable and also educates. In terms of target audience, with both series there was always the knowledge that these films would be for the wider public but also for school children and tertiary students.

This article reports on a planned evaluation, inspired by interesting responses from viewers and educators with the first series. We wanted to find out more information on the film’s effectiveness, specifically, to see if these films could be attributed to social and behavioural change. Financially, each of the series was limited to making a series of eight 5-min films, each of which was pitched as a personal story in the areas of refugees and mental health; of course, there are many more stories to tell. In the refugee series the Woven Threads team wanted to show that there have been refugees for many years so people were selected who had come to Australia over a long period of time from 1956 to most recently the Syrian war. It was important to have an equal representation of men and women, and this is reflected in the films. In the mental health series, specialists were consulted to work out what mental ill health issues would be relatable to most people. There are many more stories that need to be told, but the team had to start somewhere. They believed it is enough to watch one story but each story is about a different person’s experience of mental ill health, so if a person wanted to have a greater understanding of mental health they would be advised to watch them all. It would be desirable for people to watch all eight episodes but it isn’t necessary to have an effect or create a response.

Series 1 *Refugees: Stories from Afar* includes 8 short animated films that highlight the narratives of refugees and Series 2 *Mental Health: Stories from Within* includes 8 short animated films focused on stories of mental ill-health. Two films, described below, one for each series, were selected for the current study.

Series 1: Hisham

It’s 2014 and Hisham is 13. He lives with his family in Homs, one of Syria’s most important industrial centres. His father, not an Assad supporter, protests peacefully every day until he is shot by a sniper. When his father disappears, Hisham thinks his childhood is ended. As the oldest of his siblings and a male, he knows he has to step-up and fill the void.

Series 2: Olive

Olive is sixteen, she lives with Complex Post-Traumatic Stress Disorder (CPTSD), Major Depressive Disorder, ADHD, Autism Spectrum Disorder (ASD) and Anxiety. At the age of nine, Olive began to have negative thoughts, hidden from everyone until she was eleven and a half, the first year of high school. After much school refusal, multiple psychologists, and several schools Olive was admitted to a youth mental health unit. She tells her story, her challenges and how she grows from her experiences, not always smoothly but always forwards. Since making this film, Olive has had several difficult periods but she is growing in her self-knowledge and learning new techniques for coping with her extremely challenging life.

### 2.2. Data Collection

The description and link to the study were shared on the social media platforms [facebook, instagram, twitter, Linked In] of the participating medical research institute [withheld for blind review] as well as the Woven Threads website. We acknowledge the potential for selection bias here e.g., people who are already aware of mental health issues, younger people, people with technology access, proficiency, and time, perhaps academics, researchers and clinicians/service providers. However, given the different social platforms accessed and the age range of participants, we are confident that we achieved some diversity in our sample group. Visitors to the Woven Threads website viewed a short description about the purpose of Woven Threads (reminding us of our shared humanity), research objectives (to obtain meaningful data on the effectiveness of these films in that goal) and the opportunity to participate anonymously by viewing one or both of the two films and completing a brief Likert-type questionnaire and open-ended questions via the Qualtrics platform. The survey, including the open-ended questions, were constructed via conversations between the research team and the Woven Threads team, and also based on previous literature on the use of film as a knowledge translation strategy.

### 2.3. Analysis

Survey data was analysed using basic descriptive statistics. We used reflexive thematic analysis [13,14] to analyse the rich narratives submitted via open-ended survey responses. After careful review of the textual data, a coding frame for the interviews was developed. Themes were inductively analysed, through examining repeated patterns of meaning; variability and consistency; and commonality and differences across the sample. Strategies for ensuring research rigor included prolonged engagement with the subject matter, a team approach to analysis, an audit trail and ongoing reflexive practice [15]. This research was reviewed and approved by The University of New South Wales Human Research Ethics Committee (HC210526).

## 3. Results

### 3.1. Quantitative

364 people responded to the Refugee short film [Hisham] and 275 to the Mental Health short film [Olive]. The mean age of respondents was 51.6 years with a range from 18 to 83 years. The majority (85%) identified as Caucasian in both groups. For viewers of Olive’s story, 79% identified as female, whereas 49% of viewers of Hisham’ story identified as female. In the case of both films just under half lived in regional or rural communities and 52% living in urban settings. Over half of respondents in both groups reported that they had been diagnosed with a mental health issue (see Table 1).

The survey included a series of Likert type questions about viewer enjoyment, the film’s ability to challenge misconceptions, role in helping viewer to understand themselves or others, recognising that they or someone else might need help, and the film’s ability to evoke feelings of hopefulness. The majority (80%+) agreed that they enjoyed the films and indicated that they were a powerful way to challenge misconceptions about refugees (88%) and about individuals impacted by mental ill-health (89%) (see Table 2 and Table 3).

Respondents were asked to select one word that reflected how the film made them feel. The words that were identified were entered into a word cloud (Figure 1). The larger the font is indicative of the words that were most endorsed. In both films, the words that came to mind which were most popular immediately following viewing the film were hope/hopeful, sad/sadness, and empathy.

### 3.2. Qualitative

Qualitative narrative text from viewers of the refugee and mental health films were treated together as one data corpus due to their similar thematic content. One central overarching theme was the use of film as a knowledge translation strategy and this consisted of four central subthemes: the powerful role animated film had with respect to: (i) changing perceptions and inspiring empathy, (ii) enhancing literacy, (iii) enhancing the influence of storytelling, and, (iv) encouraging hope and a sense of belonging.

#### 3.2.1. Film as a Knowledge Translation Strategy

Respondents indicated that the characteristics of the film itself were amenable to translating knowledge about refugees and people impacted by mental health issues. They identified the accessibility and short length of the films as a positive feature, noting that people tend to desire information in small packages. The films were reported to be a powerful way to engage people and spread messages quickly. One respondent noted that information could be easily shared and sparked conversation, more so than documentaries or longer pieces.

The use of visual images in the animation, the background musical score, and the voiceover itself were viewed as central to instilling an emotional connection with the material. The medium was generally thought to be evocative and engaging. The animated nature of the visuals themselves helped to soften the harshness of the reality being told, moderating the impact of difficult material. The immediacy of impact was frequently mentioned, for example, one respondent noted that they attained fresh insights. One viewer of Hisham’s story elaborated on the role of animated film:

Removing the real-life visual aspect can actually enhance the story being told. Society has become so de-sensitised to violence and horror that using animation to convey a rather bleak subject can allow viewers to focus more on the words being spoken and the emotions of the speaker. It makes the whole thing feel more intimate, and can certainly encourage the viewer to think differently about the subject, thereby challenging preconceived notions or opinions.

Another viewer [Hisham] also noted that the voice over of the person humanizes the content of the film.

Using animation to convey a rather bleak subject can allow viewers to focus more on the words being spoken and the emotions of the speaker.

Several viewers stated the practicality and effectiveness of enhancing perspectives via the medium of short film.

I think it’s a brilliant way to shape perspectives. It’s not didactic. When something isn’t hitting you over the head with a cause, you’re more able to hear and see what’s being represented. Instead, it reveals the personal impact of a negative experience.

Short, animated films have the potential to transport the viewer in an accessible way, allowing for new perspectives/contexts. At the same time, such films can spark one’s imagination, liberating them from feelings of despair towards positive action.

It brings the words to life, creates a vividness that people can immediately relate to and this makes people feel emotions much stronger than just words. This would lead to stronger thoughts, more likely to take action based on the film.

#### 3.2.2. Changing Perceptions and Inspiring Empathy

Respondents identified the role the films had in changing their perceptions of the refugee experience and of experiences of mental ill-health. One respondent viewer of the refugee film indicated that the personal story represents an invitation into the narrative. In this way, the viewer empathizes with the individual and looks at others with compassion rather than judgment. Another respondent noted that the film would allow for more compassion towards refugees as a result of hearing stories from people who are commonly discriminated against or shown prejudice. The film was reported to encourage awareness of the backgrounds of people who were not like them. As a consequence, several respondents noted that stereotypes can be broken down.

A sense of empathy was often mentioned explicitly. Empathy refers to the capacity we share as human beings to step into each other’s shoes. It allows us to understand where another individual is coming from and what they are feeling. It involves deep listening and a refusal to be judgmental. One viewer of the mental health film stated:

This reality was brought into the open. The way this story is told by the person herself helps me to relate better to the situation and to feel closer to the issue itself. It also helps me to keep my eyes open and pay attention to people that might have mental health issues.

It can help people understand what it’s like to live with mental health issues, as too often they are ‘invisible’ and people lack understanding. It can help people to understand more, and be more empathetic.

Participants identified that the pictures, animations and music bring the videos to life. They emphasized the emotions, which allowed the viewer to see what the person went through. They were found to tell a compelling story and allowed people to empathize with the character. This was explicitly linked to an enhanced likelihood of action being taken following their viewing.

#### 3.2.3. Enhancing Literacy and Potential for Social Change

The films were reported to increase viewer knowledge, education and understanding of mental health and refugee issues and experiences. Knowledge about what it might feel like to be a refugee or to experience mental ill-health was accompanied by practical information about available resources in the community.

Although the animation is brief, much significant information is presented and important mental health issues are highlighted. The animations also suggest where to find the centres where mentally ill people can get real help.

Some participants described the ways in which they were engaged, curious and wanted to know more about mental health issues. They described the potential for using the films as an engaging educational tool in schools, camps or sessions with a group of young people who could then discuss it. One respondent noted that it could make a huge impact on young people; as an accessible animation which reduces that distance between someone experiencing depression and others is incredibly valuable. Others supported the view that the films could be used for educational purposes in schools and in the workplace:

This would be a fantastic tool to use in upper primary and high schools. It can be part of the curriculum.

My employer has been advocating for mental health in the workplace, so it is becoming less of a stigma. I can see the impact stigma has in my workplace, and we have seen great results by advocating for mental health.

For those experiencing similar issues (particularly teenagers), it illustrates that they’re not alone and provides them with terminology/reference points to describe their own experiences and seek support. For those who don’t directly relate, such short films provide a generous insight into the reality of mental illness and the crucial importance of professional intervention and support.

The film was described as bringing a great deal of hope to someone who is struggling with mental ill-health.

Respondents identified the sense of connectedness in both films which made the story more relatable particularly when one has not had such experiences. As a result of this sense of connectedness to the person’s story, many viewers spoke of having the knowledge that now allowed them to change their perspectives on issues that they had preconceived concepts about. One viewer of Hisham’s story indicated the potential for social change, stating that the film allowed for awareness of hardship and suffering and the ways in which that knowledge could encourage people to come together as a community and be more proactive in helping others. Another respondent noted that the film provided information about what people in war torn countries go through and which can then lead to reflection about prejudice towards refugees and to a better understanding of their circumstances. Another viewer spoke of the film as producing knowledge that encourages self-reflection and stated:

It reminds us that every day we can choose to make a positive difference to the lives of others. Reminds us how fortunate we are to live in a country that is not at war. They might look into different ways that they can help support refugees.

#### 3.2.4. The Power of Storytelling: Personal Stories Coming from Real People

A powerful thread that permeated the textual responses for both films was the story. Participants referred to the importance of the human voice and the very human perspective that people may not connect with in the news media. One viewer commented that the refugee film allows for profiling a person or people at the centre of the story, which helps people to see beyond the dehumanizing impact of refugees that are frequently headline news. Respondents indicated Hisham’s story as told via animation allowed them to hear and see the story without distraction from actual footage or distressing images of events. It was suggested that a person sharing their personal experience lends greater authenticity to the story as well as making the situation more realistic.

I think it can help to move people by connecting them to a story, lead them to think more compassionately in what they decide should happen to refugees (and how we might help them), and lead them to ultimately be more supportive of refugees coming to Australia.

I think this would be a great film to encourage people (who have a negative view of refugees) to see refugees from a different perspective as the themes relate to core values that we all have (importance of family, resilience). It isn’t a guilt trip story as it is someone’s real experience so no one could argue with that. Also, the hopeful ending was great but I think what made it better than the typical message in this space (I work in comms at NFP/charities) is that it shows people with extremely negative views that refugees can bring skills and talents to support the Australian community.

A personal story allows one to makes links with others even if they haven’t shared those experiences. Respondents stated that the personal story invites one into the narrative.

You care for the individual. They might think twice when they see someone acting differently, and look at them with compassion rather than judgment.

#### 3.2.5. Animated Film as Inspiring Hope and a Sense of Belonging

When viewers of the films were asked to produce one word that came to mind immediately upon viewing the film, ‘hope’ was identified in a great many cases [as seen by the size of the word in Figure 1 word clouds]. The narratives of respondents echoed this feeling of hope. They indicated that storytelling creates an emotional connection, accompanied by a feeling of hope. Participants observed that the films offer the opportunity to share main messages about refugees and people impacted by mental distress, and as a result, posited that people will not feel so alone in their experiences. Olive’s story, in particular, generated many comments that addressed the importance of generating hope. The stories were reported to inspire a sense of hope for others as indicated by a viewer of Olive’s story:

I feel a sense of hope knowing that others who have similar mental health disorders can come out of the blackhole.

Many respondents of both films linked the feeling of hope to a sense of connectedness, of not being alone in their experiences:

A feeling of connection, that there are other people out there suffering too-and there is hope.

I think that it could be used to show other young people in the same experience that they are not alone. To show them that they can get through what they are going through and that there are people who are there to support them, so hopefully viewers might reach out to get the support that they need, be it from their family or health professionals.

Many viewers identified that the films can provide a sense of hope and a sense of identification, that despite the fact that people impacted by mental ill-health or refugee status may feel like they are alone in their experiences, they are not.

## 4. Discussion

The current literature suggests that passive knowledge translation strategies such as print materials and clinical guidelines are less effective than active, more engaging strategies [16]. Our results suggest that animated video is a potentially effective knowledge translation strategy to engage diverse audiences in discussion and action regarding health and social care issues. These results are aligned with previous studies that indicate that the use of short film can support two main knowledge translation outcomes–knowledge acquisition and behaviour intentions [17]. For example, the use of video as a knowledge translation tool was explored in Burkina Faso with respect to health professional knowledge and training of dengue fever [18]. The animated video resulted in significantly higher increase in knowledge than did a journalistic video. Hebert et al. identified four key aspects important to consider for a video to be effective. They included communicating the evidence in a narrative form, selecting good communicators, constructing a visual tool that underlines the message and adjusting the message to the local context.

Most participants identified the positive impacts of film as a knowledge translation strategy, notable among these was accessibility. These findings are supported by other research that has demonstrated that animated film is typically perceived as non-threatening, familiar and accessible across age groups [19]. A scoping review of standalone animated videos suggested that animation is a powerful and efficient strategy for increasing knowledge when compared to other teaching formats [20]. The use of short video-clips can offer unique opportunities to open up conversations by enabling viewers to interact and discuss critical health and social care issues that are pivotal to the sustainability of health care systems [21].

The development of more video-based tools is recommended to transfer knowledge to professional and general public stakeholders. Future research should explore the impact of viewing such films over the longer term with a view to measuring behavioural impact and sustainability.

### Limitations

Responders accessed the short film in an uncontrolled setting (via electronic device at a time and place of their own choosing). There was the potential for a self-selecting sample via social media advertisements and subscribers to investigator organisational channels. Participants were primed to the concept of hope by a survey statement “I was left with a feeling of hopefulness after seeing the film”, which may have influenced the predominate theme of hope in the narrative text. Further, we did not measure or control for political orientation, which may have influenced the responses due to coverage of political decisions regarding refugee policy in Australia. We did achieve some [albeit limited] diversity of the sample, particularly with respect to Hisham’s story. Over half (51%) of viewers of Hisham’s story were not born in Australia compared to 15% of viewers of Olive’s story. A significant percentage of audience/responders for both films were highly-educated (34% had a post-graduate degree), although we did not collect details of the degree itself. Future research might include comparatively testing the impact of the films in targeted groups.

## 5. Conclusions

Short animated films to highlight the lived experience of refugees and of people with mental health challenges offers a lay friendly and accessible way to share stories. We argue that viewing an animated film produces instances of deep reflection wherein attitudes can be altered, and new affective reactions to difference can be created. Through self-reflective engagement with the film, some viewers come to recognize their own assumptions underpinning their perspectives of refugees, individuals impacted by mental ill-health and garner empathy for others who are impacted by refugee status and/or mental ill-health.

## Figures and Tables

**Figure 1 ijerph-19-11468-f001:**
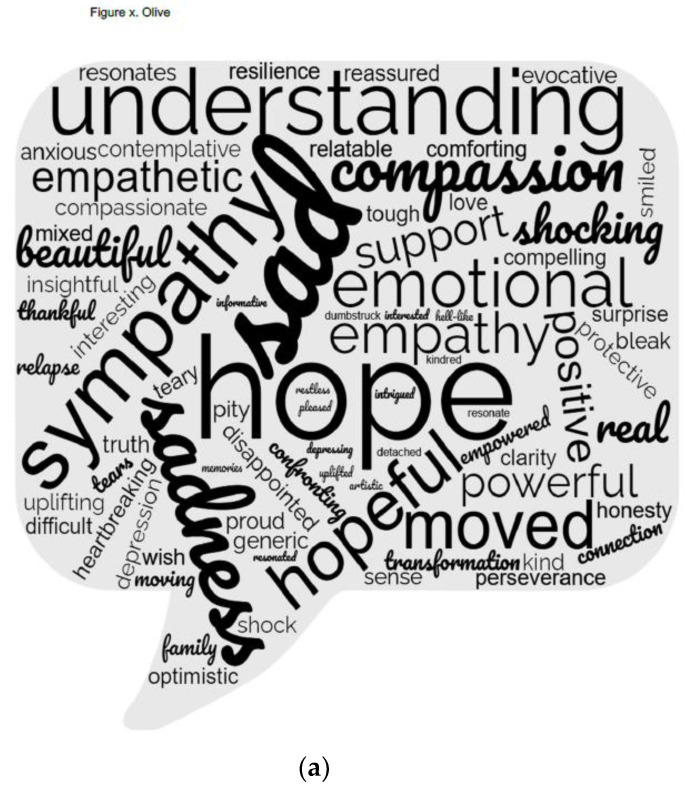
In one word, describe your immediate reaction to the film (**a**) Immediate reaction in one word to Refugee short film; (**b**) Immediate reaction in one word to Mental Health short film.

**Table 1 ijerph-19-11468-t001:** Demographic Data.

Film	#	Location	Sex	Australian Born	Education	Diagnosed Mental Health Issue	Self-Identify as Having a Mental Health Issue
Hisham’s story	374	48% regional/rural	49% female	49%	33% postgraduate	58%	45%
Olive’s story	275	48% regional/rural	76% female	85%	35% postgraduate	56%	44%

**Table 2 ijerph-19-11468-t002:** Viewer responses to Woven Threads Refugee short film (*n* = 364).

Woven Threads Short Film: Refugee Short Film Responses	Agreed/Strongly Agreed
This film is a helpful way to challenge public misconceptions about refugees that lead to stigma and discrimination	88%
I enjoyed watching the short film	80%
Watching the film helped me understand myself or someone else better	70%
I was left with a feeling of hopefulness after seeing the film	64%
After seeing the film, I recognise that I or someone else might need help	47%

**Table 3 ijerph-19-11468-t003:** Viewer responses to Woven Threads Mental Health short film (*n* = 275).

Woven Threads Short Film: Mental Health Short Film Responses	Agreed/Strongly Agreed
This film is a helpful way to challenge public misconceptions about mental health that leads to stigma and discrimination	89%
I enjoyed watching the short film	83%
Watching the film helped me understand myself or someone else better	82%
After seeing the film, I recognise that I or someone else might need help	70%
I was left with a feeling of hopefulness after seeing the film	66%

## Data Availability

Not applicable.

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
