# Peer review of "A Creative Approach to Knowledge Translation: The Use of Short Animated Film to Share Stories of Refugees and Mental Health"

_ijerph, 2022, doi:10.3390/ijerph191811468_

Round 1

Reviewer 1 Report

This is an important study into short animations as a means to educate the public about lived experiences of refugees and people with mental health conditions. The project has a lot of potential for scaling up and broader application to other topics. In order to reach this full potential, the manuscript needs improvements especially to clarify the project and make it more appraisable, reproducible and applicable by the readers. 

Disclaimer: The manuscript was obtained unblinded; from the omission of the institution's name in the manuscript it seems that the authors were expecting a blinded review. The manuscript was also obtained already in a journal article format and this reviewer cannot ascertain whether some typos and word cloud-related issues were from the original submission or occured during the formatting stage by the journal. 

Abstract

Overall: It is unclear whether this was supposed to be a structured on unstructured abstract. "Results" was clearly indicated but no other parts in the Abstract were. Please make it consistent either by removing "Results" or signposting all parts of the Abstract.

Line 13: "Mixed-methods exploratory design" was only mentioned in the Abstract and listed as a keyword, but neither mentioned nor elaborated in the body of the manuscript. In this revierwer's perspective this is not a mixed-methods study but a multi-methods study, because the quantitative and qualitative findings were neither used to inform each other nor synthesised to produce more meaning than a simple sum of both components. The authors are strongly encouraged to review the literature about the what mixed-methods research is and is not. If the authors believe that this is a true mixed-methods study, then the justification needs to be provided. At any rate, the study design must be described with proper references in the Materials and Methods section. 

Introduction

Line 29: Acknowledging different models of KT, it seems that the authors only focused on the translation of research findings into practice which in more complex models (e.g. https://ktdrr.org/ktlibrary/articles_pubs/ktmodels/figures1-6.html#figure1cih) is one of several steps in KT. This approach is not a weakness per se but needs to be explicitly mentioned. As will be indicated below, there were opportunities within this study which may or may not engage other KT steps. 

Line 74-98: The Woven Threads Series should be reported in Materials and Methods because it provides the backdrop for the whole project. Much more information is needed, both for the series overall and for the two chosen videos, so this manuscript could be properly appraised and the project could be reproduced/applied elsewhere. Who made the series? Who made the movies -- students/professionals/clinicians? Why the two stream topics, and what was the development process -- was there any consultations with the community at large, the target group, and people with lived experiences? Who was the target audience? Has it been evaluated before? Where does this project sit in WT project? Was it a planned evaluation, part of a more comprehensive evaluation, a pilot, other? What's the rationale of each movie choice in each stream, and not others? How long is each story? How was each pitched? Is it enough to watch one story to change somebody's understanding? Why not watch all 8 in the series?  

Line 78: "Understanding, in turn, is key to social change" -- Is this assumed, or has been proven in the literature? Consider the complex context of social change where undertanding alone is unlikely to result in social change. 

Line 99: As per previous comment in Abstract, study design must be stated and its rationale justified here. There are strong indications that this is a multi-methods study and not a mixed-methods study. 

Line 100: How was the survey (both quantitative and qualitative) constructed? What were the questions used especially for the qualitative part? How were those questions developed? What was done to establish the survey's validity and reliability? Was it piloted and tweaked? Did it use standardised questions? 

Line 101: Who are the institution's social media followers? Discuss selection bias here e.g. people who are already aware of MH issues, younger people, people with technology access, proficiency, and time, perhaps academics, researchers and clinicians/service providers. It is insufficient to just briefly mentioning it at the end of the paper under Limitations. 

Line 112: Why is transcription needed when data were already collected as text?

Lines 115-116: These strategies need elaboration beyond just naming/listing them, to enable critical appraisal of this study. 

Table 1: Educational level needs more elaboration beyond postgrad yes/no. 

Figure 1: For ease of access to all readers, the words should be in larger font overall so even the smallest word can still be read, and please do not use colours because they may disadvantage people with colour-blindness. The second word cloud being in two separate geometric shapes gives the impression there were two separate word clouds for Olive's story. Please make it into just one shape. 

Line 156-157: The stem for the subtheme creates a sentence structure which doesn't work for subtheme (iii): "The powerful role animated film had to the power of storytelling".

Lines 159-326:

a. The whole qualitative results section is very difficult to read and understand because the quotes are not presented in a particular way to differentiate them from the rest of the text. Quotation marks are used sparingly and inconsistently throughout the section.  

b. The numbering is also confusing; it reverts back to 1, 2, 3 after going down to 3.2.1.

c. Some definitions e.g. lines 205-208 need to be referenced. 

d. Some parts, especially when referring to participants' lived/shared experiences, indicate that the qualitative analysis may have been linked to participants' demographic data. If this is the case, then this linkage should be reported, and each quote should be labeled e.g. "...." (male with lived experience). 

e. Lines 254-255 indicate "the ways in which both films could make the story more relatable particularly when one has not had such experiences." These ways need to be reported so the readers could benefit from them.

f. Lines 299-301 state the emergence of "hope" as a theme; but the authors failed to acknowledge that participants had been primed to this concept by a survey statement "I was left with a feeling of hopefulness after seeing the film". This survey item unfortunately has 'contaminated' the participants' subsequent free text answer by planting that concept as a theme. This problem cannot be rectified but at the very least must be acknowledged because its potential bias cannot be dismissed.

Discussion

Lines 328-347: This section is incredibly limited, and sadly does not convey anything new for the body f knowledge. It needs much more elaboration and there are many ways to do this e.g. the Australian context, other KT levels etc. 

Line 354: The claim of achieving "diversity of the sample" is too broad when only very limited demographic data were (collected and) presented. Refugee experience, socio-economic status, and residential postcode are only a few examples of factors which may influence participants' veiws and perspectives -- not just political leanings which the authors correctly identified.

Conclusion 

It is unclear where "viewers come to recognize assumptions" and "the vulnerability of their own embodied lives" came from; these aspects are not supported by the reported data. While some participants did share their MH diagnosis and self-assessment in the survey, data on participants' refugee experiences did not seem to be collected, and therefore only a proportion of participants can be claimed to have "embodied lives" in relation to the two movies. 

General comment: Pleae proofread and correct all typos in the manuscript. 

Author Response

Thank you for taking the time to comment on our paper submission. Our response to the review is detailed in the attached document.

Reviewer 2 Report

Dear authors

I had an enjoyable time reading your work, however, in order to make this work suitable for publication you have to address some issues in your manuscript. The following paragraphs include my main concerns. 

Abstract: you mention your study adopted a mixed-method exploratory design, however, only qualitative findings are available in this section. Please, include your quantitative findings as well and underline the overall implications of both strands. 

Introduction: the authors spent a lot of time about the different tools available to facilitate learning and catalyze behavior change. However, as a reader, I want to know why this study is worthy of reading. Moreover, how this work would contribute to making knowledge available and useful for practitioners, scholars, and policymakers. Also, it would be wise to inform the readers about the struggles that scholars face in achieving knowledge translation, and its current and future negative effects.  

Method: The authors must explain why they adopted a Mixed-method exploratory design. As Creswell & Pliano (2014), I think the adoption of this approach should not respond to the personal desires of researchers but to the research needs. Therefore, the authors must explain why using either quantitative or qualitative approaches would not add to the field. Moreover, as far as I know, the two forms of data must be integrated into the design analysis through merging the data, connecting the data, or embedding the data (see Creswell and Plano, 2014; Guest and Fleming, 2015). To be honest, considering your study in its current form, I see your study as a qualitative study, not a mixed-method approach. 

Also, there are three main types of mixed methods designs (convergent, exploratory, and explanatory). I have serious doubts that the type you adopted is appropriate for your study, considering the data collection process and the analyses as well. Please, revisit this section carefully. 

Analyses: Please provide further explanation about this process. That is how your quantitative findings informed your qualitative analyses and/or vice versa. 

Results/Findings: It is hard to read this section; even though the authors say they used thematic analysis to analyze data this section is lack excerpts that may illustrate their findings. You may want to add some excerpts to illustrate what you report.

Discussion/Conclusion: It is unclear in what ways this work is different from previous research, overall how a mixed-method study would contribute to closing the current gaps in the literature. More importantly, the authors should underline the practical and theoretical implications of their findings and how those will shed light on a long-lasting issue of knowledge translation.

Author Response

(The authors gave the same response as above.)

Round 2

Reviewer 1 Report

Thank you for accommodating my inputs in the revised paper, which is now more informative and hopefully would create interest for other researchers to build on.